# Association between Workers’ Anxiety over Technological Automation and Sleep Disturbance: Results from a Nationally Representative Survey

**DOI:** 10.3390/ijerph191610051

**Published:** 2022-08-15

**Authors:** Seong-Uk Baek, Jin-Ha Yoon, Jong-Uk Won

**Affiliations:** 1Department of Occupational and Environmental Medicine, Severance Hospital, Yonsei University College of Medicine, Seoul 03722, Korea; 2The Institute for Occupational Health, Yonsei University College of Medicine, Seoul 3722, Korea; 3Graduate School, Yonsei University College of Medicine, Seoul 03722, Korea; 4Graduate School of Public Health, Yonsei University College of Medicine, Seoul 03722, Korea; 5Department of Preventive Medicine, Yonsei University College of Medicine, Seoul 03722, Korea

**Keywords:** automation anxiety, sleep disturbance, worker

## Abstract

Despite the positive aspects of recent technological innovations, fears are mounting among workers that machines will inevitably replace most human jobs in the future. This study is the first to explore the association between individual-level automation anxiety and insomnia among workers. We scored the worker’s anxiety over technological automation with five questions. The total sum of scores for participants was categorized in quartiles (Q1–Q4). Logistic regression was employed to estimate odds ratios (ORs) and confidence intervals (CIs). The highest scoring group (Q4) had the highest OR for sleep disturbance (OR [95% CI]:1.40 [1.27–1.55]) compared to the lowest scoring group (Q1). ORs of the highest scoring group (Q4) were strongest for the young (OR [95% CI]:1.96 [1.52–2.53]), followed by the middle-aged (OR [95% CI]:1.40 [1.20–1.64]), and old age groups (OR [95% CI]:1.29 [1.10–1.51]). In addition, a 1-point increase in the automation anxiety score had the strongest association with sleep disturbance in the young (OR [95% CI]:1.07 [1.05–1.10]), followed by the middle-aged (OR [95% CI]:1.03 [1.02–1.04]), and old age groups (OR [95% CI]:1.02 [1.01–1.04]). Our study suggests that policies such as worker retraining are needed to alleviate workers’ undue anxiety.

## 1. Introduction

In the era of the Fourth Industry Revolution, rapid technological advances have brought considerable changes to the structure of industries and occupational health and safety [1]. For example, the increase in work arrangements through digital platforms or the globalization of companies is expected to weaken regulations on occupational safety and health [2]. Technological innovations have not only increased productivity but also contributed to improving worker safety and health by preventing dangerous situations or reducing physical workload [3,4]. Despite the positive aspects of these recent innovations, fears are mounting among workers that machines will inevitably replace most human jobs in the future [5]. In recent decades, the adoption of automation technologies has steadily increased, leading to an increase in unemployment or a decrease in the wages of workers [6]. According to one previous study, cashiers, legal secretaries, and accounting clerks might be replaced by automation with a 97–98% chance [7]. This trend has been reinforced by the recent progress in artificial intelligence (AI), which has enabled human-robot interactions in various industrial fields [8,9]. Therefore, the concept of ‘automation risk’, which means the probability of occupations or tasks being replaced by automation, has drawn attention over the past decade [7]. Automation, which used to threaten only blue collar workers, is now causing job insecurity for most service and sales workers or even highly skilled white collar workers [10].

While most previous studies on workplace automation have investigated its impact on the labor market, few studies have explored the possible health outcomes of technological automation. While some researchers predicted that automation would improve workers’ health by replacing physically demanding tasks, the introduction of automation was reported to induce an increase in mortality rate by causing unemployment [11]. In addition, according to one mediation analysis, automation risk results in deterioration of subjective health by increasing perceived job insecurity [12]. Erebak and Turgut revealed that workers’ perceived job insecurity increases as they become more anxious about the speed of technological development and automation [13]. Job insecurity is well known as the cause of workers’ poor mental health [14]. An association between workers’ concern over their jobs being replaced by smart machines and job dissatisfaction has also been established in the literature [15]. Conversely, the positive health impact of automation has been suggested by previous studies, where workers facing automation risk reported less psychological stress [16] and burnout [17]. However, the majority of studies estimated automation risk according to workers’ occupations, and none of the studies have measured individual perceptions of technological automation.

Furthermore, workers react differently to job insecurity and precarious employment depending on their age [18,19]. Similarly, the impact of workers’ anxiety over automation could vary according to age. Preceding studies found that not only are young workers under the age of 40 more afraid of losing their jobs due to automation [20], but their mental health is also more affected by anxiety over automation [21].

Insomnia experienced by workers not only deteriorates their health but also adversely affects organizations and society by reducing work performance [22]. Workers’ sleep disturbances can be induced by multiple occupational factors, including shift work [23] and employment type [24]. Moreover, psychological factors are widely known as important determinants of workers’ sleep quality [25]. For instance, previous studies reveal that job stress, job satisfaction, and facing complaining customers are related to workers’ sleep disturbances [26,27]. In addition, perceived job insecurity and precarious employment are related to decreased sleep quality [28,29].

Consequently, it is reasonable to hypothesize that workers’ negative perceptions of technological automation may be related to sleep disturbance and that younger workers may be more vulnerable to the detrimental effects of automation anxiety. This study is the first to explore the association between individual-level automation anxiety and insomnia among workers. We hope that our study findings will further expand the current knowledge of the effects of technological automation of the Fourth Industrial Revolution on workers’ health.

## 2. Materials and Methods

### 2.1. Study Sample

The sixth Korean Working Conditions Survey (KWCS) was used in this study, which was conducted from October 2020 to April 2021. The nationwide KWCS has been conducted every three or four years since 2006 by the Occupational Safety and Health Research Institute (OSHRI) of Korea. The research items of the KWCS benchmarked those of the European Working Condition Survey (EWCS) conducted by Eurofound, aiming to collect data on working conditions that affect the health and safety of workers. Participants aged >15 years were selected by multistage, stratified, random sampling. A trained interviewer conducted a one-on-one interview with each participant.

Of the initial 50,538 participants, we limited participants to workers aged over 18 years, leading to 50,493 adult participants. Next, we excluded 3968 participants with missing values. A final total of 46,525 workers were included in this study. The flowchart of the study sample selection is presented in Appendix A.

### 2.2. Data Availability and Ethics Statement

The raw KWCS data does not contain personal information and is openly published online (https://www.kosha.or.kr/eoshri/index.do, accessed on 1 July 2022). The Institutional Review Board of Severance Hospital approved this study (approval number: 4-2022-0507).

### 2.3. Main Variable

The items regarding the worker’s perception of technological automation were translated and newly included in the sixth KWCS with reference to the seventh EWCS [30]. For each wave of the EWCS, the development of the questionnaire is subject to a thorough review process by the expert questionnaire development group composed of representatives of labor unions, survey institutes, researchers, international organizations, and agencies. Furthermore, to verify the validity of the new questions, a pre-test is performed prior to finalization [31]. Since technological automation has recently become an emerging issue, the questions regarding automation anxiety were introduced in the seventh EWCS.

Automation anxiety (“How concerned are you about the following situations in which technological advances and automation can affect your work in the future?”) was assessed with the following five items: (i) decrease in control over how it is performed, (ii) difficulty in using skills and ability, (iii) decrease in income, (iv) being transferred to an uninteresting job, and (v) unexpected changes to working hours. Each item was measured as: (0) “Not at all concerned,” (1) “Not very concerned,” (2) “Fairly concerned,”, and (3) “Very concerned.” Then, scores were summed such that a higher total score indicated a higher level of anxiety over technological development and automation (range: 0–15). Cronbach’s alpha was used to measure internal consistency, which was 0.89. The total sum of scores for participants was categorized in quartiles (Quartile 1–Quartile 4).

Sleep disturbance was assessed with the Minimal Insomnia Symptom Scale (MISS) [32], which consists of the following three sleep-related difficulties: (A) difficulty falling asleep, (B) night awakening, and (C) not being refreshed after sleep. Possible responses are (0) “never,” (1) “rarely,” (2) “several times a month,” (3) “several times a week,” and (4) “daily.” The total score ranges from 0 to 12, and based on previous research, those with ≥6 points were defined as having sleep disturbances [32].

### 2.4. Covariates

Age group, gender, education, monthly income, occupation, working hours, employment status, shift work, job stress, job satisfaction, and dealing with angry customers were considered as possible confounders. As multiple meta-analyses and review articles consistently report that insomnia of workers is influenced by psychological factors such as job stress, job satisfaction, and emotional job demands, we regarded them as possible confounders [25,33,34]. Moreover, previous studies suggested that emotional job demands such as handling clients could act as a confounder in the relationship between automation risk and mental health [17,35]. Age groups were categorized as ≤35 years (young), 36–55 years (middle-aged), and ≥55 years (old). Our study samples consisted of 21,833 men (46.9%) and 24,692 women (53.1%). Education level was categorized as having completed middle school or lower, high school, or college or higher. Monthly average income was categorized as <2,000,000 won; 2,000,000–2,990,000 won; 3,000,000–3,990,000 won; and ≥4,000,000 won. Occupation was classified as blue-collar work, service and sales work, and white-collar work, based on the Korean Standard Classification of Occupations. Working hours per week were categorized as ≤40 h, 41–52 h, and ≥53 h. Employment status was categorized as permanent, temporary/daily, self-employed, and unpaid family work. Regarding job stress, participants were asked “Do you experience stress in your work?” According to the answer, participants were categorized as “low” (never, rarely), “middle” (sometimes), and “high” (most of the time, always). Regarding job satisfaction, participants were asked “On the whole, are you very satisfied, satisfied, not very satisfied, or not at all satisfied with the working conditions in your job?” According to their answers, participants were categorized as “low” (not very satisfied, not at all satisfied) and “high” (very satisfied, satisfied). Finally, participants were asked: “Does your job involve dealing with angry clients?”. Participants were categorized as “rarely” (never, almost never), “sometimes” (1/4 of working hours, half of working hours), and “always” (3/4 of working hours, almost always, always).

### 2.5. Statistical Analysis

For descriptive analyses, the baseline characteristics and prevalence of insomnia among the study samples were analyzed using the χ^2^ test. In addition, scores for automation anxiety according to the characteristics of the study samples were analyzed using either ANOVA or Student’s *t*-test. For regression analyses, we used logistic regression with survey weights to estimate the odds ratios (ORs) and 95% confidence intervals (CIs). We transformed regression coefficients into ORs to make our results more interpretable. The OR represents the strength of the association between exposure (automation anxiety) and outcome (insomnia) [36]. First, the total score of automation anxiety was included in logistic regression either as a categorical (Q1–Q4) or continuous (range: 0–15) variable. The interactional effect of each age group and automation anxiety on insomnia was explored by including interaction terms. Finally, subgroup analyses by age group were performed to investigate the differential impact of automation anxiety on insomnia. All statistical analyses and visualization were performed using R (version 4.2.0; R Foundation for Statistical Computing, Vienna, Austria).

## 3. Results

Table 1 presents the sociodemographic features and prevalence of insomnia among study participants according to each quartile of automation anxiety. Of the total 46,525 workers, 4618 (9.9%) had sleep disturbance according to the MISS classification. A higher prevalence of insomnia was associated with workers who were of older age (13.9%), women (11.6%), and with lower education (19.2%) and income level (13.9%). Psychosocial occupational factors, including high job stress (13.9%), low job satisfaction (20.1%), and frequently dealing with angry customers (24.1%) were significantly related to sleep disturbance. Regarding participants’ anxiety over technological automation, the higher the score, the higher the prevalence of insomnia (Q1: 9.6% vs. Q4: 10.7%).

Table 2 compares the characteristics of workers according to each quartile of automation anxiety levels. As the table shows, the proportion of young and middle-aged workers was higher in the group with higher automation anxiety scores. In addition, workers with higher education levels and job stress, white-collar workers, and permanent employees accounted for a larger percentage of the group with higher automation anxiety levels. Among the occupational factors, the proportions of shift workers did not show significant differences between groups (Q1: 7.0% vs. Q4: 7.2%)

Appendix A indicates that young and middle-aged workers, men, workers of higher education level, service/sales workers, and white-collar workers experienced higher levels of automation anxiety. Higher levels of automation anxiety were associated with higher job stress and lower job satisfaction.

Figure 1 shows the prevalence of concern regarding each of the five situations that might result from technological automation in the future. Approximately 30–50% of workers were either fairly or very concerned about all five situations. In particular, workers’ anxiety about a decrease in income was evident, with 38.1% and 25.1% of workers being fairly or very concerned, respectively.

Results for the association between the overall level of automation anxiety and sleep disturbance are presented in Table 3. In the fully adjusted model A, the highest scoring group (Q4) had the highest OR for sleep disturbance (OR [95% CI]:1.40 [1.27–1.55]) compared to the lowest scoring group (Q1). Model C shows that a 1-point increase in automation anxiety score is associated with an increased OR for sleep disturbance (OR [95% CI]:1.03 [1.02–1.04]). Finally, for the association between automation anxiety and sleep disturbance, the moderating effect of the age group was analyzed through fully adjusted models B and D, which included interaction terms. Statistically significant negative interactions were observed for middle-aged and older workers. As Models B and D suggested, an interaction effect between age and automation anxiety on sleep disturbance was shown. Figure 2 shows the predicted probabilities of sleep disturbance for each of the three age groups. Younger workers had a lower risk of sleep disturbance than older workers; however, the effect of automation anxiety was greater.

Table 4 shows the effect of worker age on the association between automation anxiety and sleep disturbance. The relationship between the highest scoring group (Q4) and sleep disturbance was strongest for the young (OR [95% CI]:1.96 [1.52–2.53]), followed by the middle-aged (OR [95% CI]:1.40 [1.20–1.64]), and old age groups (OR [95% CI]:1.29 [1.10–1.51]). In addition, a 1-point increase in the automation anxiety score had the strongest association with sleep disturbance in the young (OR [95% CI]:1.07 [1.05–1.10]), followed by the middle-aged (OR [95% CI]:1.03 [1.02–1.04]), and old age groups (OR [95% CI]:1.02 [1.01–1.04]).

## 4. Discussion

Our results suggest, for the first time, that automation anxiety is related to sleep disturbance in workers. This association remained significant after adjusting for sociodemographic features, various working conditions, and psychological demands. Furthermore, the higher the automation anxiety score, the greater the impact.

The findings from our study are consistent with preceding research, which maintained that automation risk poses a threat to workers’ health [11,12,37]. Previous studies found that the introduction of industrial robots causes workers’ unemployment or perceived job insecurity, which in turn can lead to an increase in mortality and physical or psychological distress at the county level [6,11,12,35]. Some studies have investigated the relationship between regional-level automation risk and health [11,12]. Among these, one study found that a county-level increase in the number of industrial robots was associated with all-cause mortality [11]. Another study suggested that regions with a higher proportion of occupations at high risk of automation have lower levels of health [12]. A different group of studies has explored various health conditions of workers in occupations with high probabilities of being replaced by automation. Workers in occupations vulnerable to technological replacement are associated with poor overall health [16,17], work-related injury or disease [35], and increased disability and mortality risk [37].

However, the main limitation of previous studies on the health effects of automation is that the actual level of anxiety or insecurity felt by each individual was not considered because automation risk was measured according to occupational classification using the Frey-Osborne Index [7]. The former approach assumes that workers of the same occupation have the same automation risk despite differences in age, education level, and job competence. In contrast, our study analyzed the association between individual-level automation anxiety and possible negative health effects.

The relationship between automation anxiety and sleep disturbance observed in the current study could be explained by the perceived job insecurity of the workers. Previous literature on job insecurity has mainly dealt with precarious employment status or an individual’s perception of the termination of the current labor contract [38]. However, by measuring anxiety about possible situations that will result from technological advances in the future, our variable reflects the individual’s perception of job insecurity in the next few years or decades. In this regard, our finding that automation anxiety scores were higher for permanent employees than for temporary employees reveals that job insecurity caused by automation is distinguished from the workers’ current status of employment instability (Appendix A). The negative perception of one’s job security and prospects could spill over to the home domain, impairing the quality of rest and sleep [39]. From this perspective, some studies have found detrimental effects of job insecurity on workers’ sleep quality in both Western and Asian countries [29,40].

Our study findings are also novel in that it is contrary to previous studies showing that occupations with high automation risk are related to a decrease in burnout and job stress [16,17], which are well-known factors causing sleep disorders [33,41]. As Cheng et al. suggested, these previous results may be attributable to the exposure of workers in occupations with high automation risk to lower occupation-related emotional demands. On the other hand, our study, which adjusted for psychosocial work demands and occupations, showed that individual high automation anxiety was related to sleep disturbance.

Our analyses of the age difference in the automation effect show that younger workers are most vulnerable to poor sleep quality caused by automation anxiety. This result corroborates the prior study by revealing that young workers are most affected by the detrimental mental health effects of technological automation [21]. Since it has been reported that older workers are less familiar with and sensitive to technological innovations [42], it could be assumed that they may feel relatively less anxiety about future changes in working environments due to automation. On the other hand, young workers tend to appreciate the potential of technological advancement and adapt better to it [43], yet they are more aware of the risks it poses to the labor market, and therefore the impact on sleep quality may also be greater. In fact, it is expected that in the next 20 years, human labor will be replaced by machines in a wide range of industrial sectors owing to the Fourth Industrial Revolution [2]. In this AI-driven computerization process, young workers are more likely to experience unemployment and job changes in the future as they have decades or longer remaining of their careers. For this reason, young workers may be more susceptible to health impacts at the same level of automation anxiety.

As the Fourth Industrial Revolution brings about changes in the traditional job environments, a new strategy is needed to protect the safety and health of workers [44]. Leso et al. argued that in the era of Industry 4.0, physically hazardous tasks may decrease while psychological risks increase. These psychological threats include privacy violations, job insecurity, and reduced interactions between humans [1]. Min et al. mentioned that the increase in perceived job insecurity due to automation can lead to the deterioration of workers’ mental health [2]. Our study is meaningful in that it is the first empirical study to suggest that anxiety over automation can actually worsen the quality of sleep for workers.

Our study has certain limitations. First, our findings are based on cross-sectional survey data. Therefore, further studies with a longitudinal design are needed to determine a causal relationship between automation anxiety and sleep disturbance. Next, there is a possibility that unobserved individual traits or personalities might confound the association between automation anxiety and insomnia. For example, participants with a higher level of neuroticism, who tend to respond to risks or unstable situations with negative emotions, are more likely to experience sleep disturbance [45] and feel anxious about technological advancements [46].

Despite these limitations, our study has several strengths. First, the association between automation anxiety and sleep disturbance was derived from a nationally representative sample. The KWCS also contains a wide range of variables related to the working environment and job characteristics. Therefore, one of our findings’ strengths lies in generalizability. Second, to the best of our knowledge, this is the first study to investigate the relationship between individual-level automation anxiety and health. Thus, our results support previous findings suggesting that occupational-level automation probabilities might negatively affect workers’ general health while contradicting prior findings suggesting that high automation risk is associated with reduced burnout and work stress. Finally, our study suggests that age could act as an effect modifier in the relationship between automation anxiety and sleep disturbance.

## 5. Conclusions

Policymakers and occupational health professionals should consider that workers’ anxiety over technological advances is associated with sleep disturbance. In the era of the Fourth Industrial Revolution, where the ability to manage computers and automated devices is important, more skilled workers will be needed. One previous study revealed that labor market training mitigates the negative effect of automation risk on the probability of job finding [47]. In this respect, effective worker retraining may reduce automation anxiety. Our study suggests that policies to alleviate workers’ undue anxiety can be effective in improving their sleep quality and health.

## Figures and Tables

**Figure 1 ijerph-19-10051-f001:**
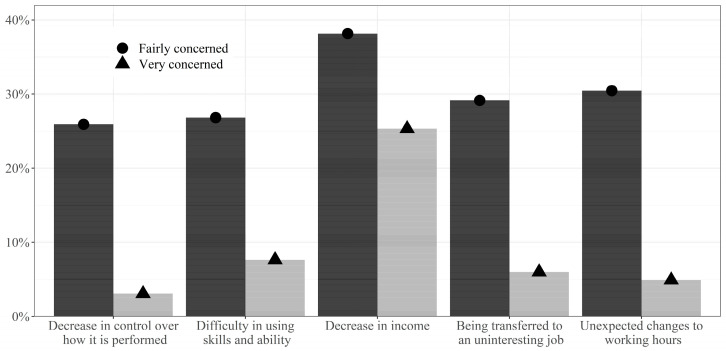
Prevalence of concern regarding each situation that may result from technological automation in the future.

**Figure 2 ijerph-19-10051-f002:**
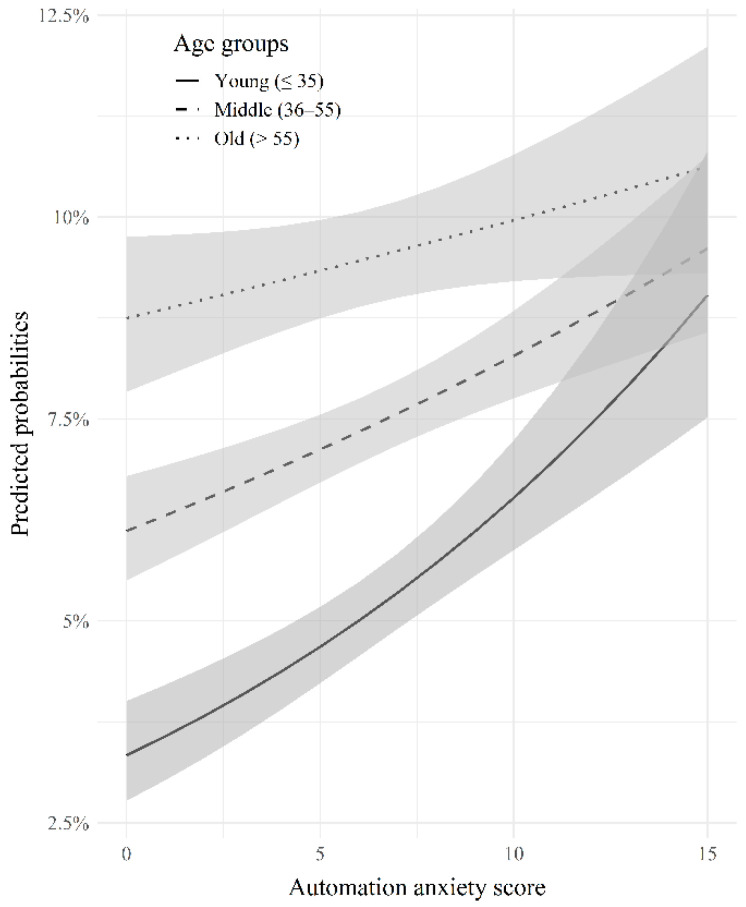
Average predicted probabilities for sleep disturbance of three age groups. Predictions were based on multivariate logistic regression with interaction terms (Model D).

**Table 1 ijerph-19-10051-t001:** Baseline characteristics and prevalence of sleep disturbance among study samples.

Characteristics	Total (*n* = 46,525)	Sleep Disturbance		*p* Value ^a^
		Yes (*n* = 4618)	No (*n* = 41,907)	
Automation Anxiety (Categorical)				
Q1 (Lowest)	11,904 (25.6%)	1144 (9.6%)	10,760 (90.4%)	0.037
Q2 (Lower middle)	12,695 (27.3%)	1229 (9.7%)	11,466 (90.3%)	
Q3 (Higher middle)	11,452 (24.6%)	1129 (9.9%)	10,323 (90.1%)	
Q4 (Highest)	10,474 (22.5%)	1116 (10.7%)	9358 (89.3%)	
Age groups (years)				
Young (≤35)	8838 (19.0%)	546 (6.2%)	8292 (93.8%)	<0.001
Middle-aged (36–55)	20,641 (44.4%)	1705 (8.3%)	18,936 (91.7%)	
Old (>55)	17,046 (36.6%)	2367 (13.9%)	14,679 (86.1%)	
Gender				
Men	21,833 (46.9%)	1762 (8.1%)	20,071 (91.9%)	<0.001
Women	24,692 (53.1%)	2856 (11.6%)	21,836 (88.4%)	
Education				
Middle school or below	7846 (16.9%)	1507 (19.2%)	6339 (80.8%)	<0.001
High school	17,237 (37.0%)	1485 (8.6%)	15,752 (91.4%)	
College or higher	21,442 (46.1%)	1626 (7.6%)	19,816 (92.4%)	
Monthly income (1000 ₩)				
≤2000	15,652 (33.6%)	2169 (13.9%)	13,483 (86.1%)	<0.001
2000–2990	14,405 (31.0%)	1214 (8.4%)	13,191 (91.6%)	
3000–3990	9309 (20.0%)	652 (7.0%)	8657 (93.0%)	
≥4000	7159 (15.4%)	583 (8.1%)	6576 (91.9%)	
Occupation				
Blue collar	17,013 (36.6%)	2074 (12.2%)	14,939 (87.8%)	<0.001
Service/sales worker	14,004 (30.1%)	1292 (9.2%)	12,712 (90.8%)	
White collar	15,508 (33.3%)	1252 (8.1%)	14,256 (91.9%)	
Weekly working hours				
≤40	28,292 (60.8%)	2074 (12.2%)	14,939 (87.8%)	0.068
41–52	10,732 (23.1%)	1292 (9.2%)	12,712 (90.8%)	
>52	7501 (16.1%)	1252 (8.1%)	14,256 (91.9%)	
Employment type				
Permanent	23,643 (50.8%)	1868 (7.9%)	21,775 (92.1%)	<0.001
Temporary/daily	7115 (15.3%)	847 (11.9%)	6268 (88.1%)	
Self-employed	14,395 (30.9%)	1691 (11.7%)	12,704 (88.3%)	
Others	1372 (2.9%)	212 (15.5%)	1160 (84.5%)	
Shift work				
No	43,202 (92.9%)	4246 (9.8%)	38,956 (90.2%)	<0.001
Yes	3323 (7.1%)	372 (11.2%)	2951 (88.8%)	
Work stress				
Low	11,457 (24.6%)	1005 (8.8%)	10,452 (91.2%)	<0.001
Middle	21,450 (46.1%)	1724 (8.0%)	19,726 (92.0%)	
High	13,618 (29.3%)	1889 (13.9%)	11,729 (86.1%)	
Job satisfaction				
Low	7989 (7.8%)	1608 (20.1%)	6381 (79.9%)	<0.001
High	38,536 (82.8%)	3010 (7.8%)	35,526 (92.2%)	
Facing angry customers				
Rarely	39,290 (84.4%)	3465 (8.8%)	35,825 (91.2%)	<0.001
Sometimes	5580 (12.0%)	754 (13.5%)	4826 (86.5%)	
Always	1655 (3.6%)	399 (24.1%)	1256 (75.9%)	

^a^ Chi-squared test.

**Table 2 ijerph-19-10051-t002:** Baseline characteristics and prevalence of sleep disturbance among study samples.

	Automation Anxiety (in Quartile)				
Characteristics	Q1 (Lowest) *n* = 11,904	Q2 (Lower Middle) *n* = 12,695	Q3 (Higher Middle) *n* = 11,452	Q4 (Highest) *n* = 10,474	*p* Value ^a^
Automation anxiety score (mean ± SD, range: 0–15)	1.7 ± 1.5	5.4 ± 0.5	7.8 ± 0.8	11.3 ± 1.5	<0.001
Age groups (years)					
Young (≤35)	2068 (17.4%)	2394 (18.9%)	2233 (19.5%)	2143 (20.5%)	<0.001
Middle-aged (36–55)	4720 (39.7%)	5587 (44.0%)	5174 (45.2%)	5160 (49.3%)	
Old (>55)	5116 (43.0%)	4714 (37.1%)	4045 (35.3%)	3171 (30.3%)	
Gender					
Men	6534 (54.9%)	6579 (51.8%)	6259 (54.7%)	5320 (50.8%)	<0.001
Women	5370 (45.1%)	6116 (48.2%)	5193 (45.3%)	5154 (49.2%)	
Education					
Middle school or below	2774 (23.3%)	2271 (17.9%)	1688 (14.7%)	1113 (10.6%)	<0.001
High school	4271 (35.9%)	4657 (36.7%)	4487 (39.2%)	3822 (36.5%)	
College or higher	4859 (40.8%)	5767 (45.4%)	5277 (46.1%)	5539 (52.9%)	
Monthly income (1000 ₩)					
≤2000	4972 (41.8%)	4463 (35.2%)	3583 (31.3%)	2634 (25.1%)	<0.001
2000–2990	3231 (27.1%)	3798 (29.9%)	3810 (33.3%)	3566 (34.0%)	
3000–3990	2017 (16.9%)	2437 (19.2%)	2357 (20.6%)	2498 (23.8%)	
≥4000	1684 (14.1%)	1997 (15.7%)	1702 (14.9%)	1776 (17.0%)	
Occupation					
Blue-collar	4943 (41.5%)	4805 (37.8%)	4036 (35.2%)	3229 (30.8%)	<0.001
Service/sales worker	3329 (28.0%)	3553 (28.0%)	3747 (32.7%)	3375 (32.2%)	
White-collar	3632 (30.5%)	4337 (34.2%)	3669 (32.0%)	3870 (36.9%)	
Weekly working hours					
≤40	7669 (64.4%)	8108 (63.9%)	6552 (57.2%)	5963 (56.9%)	<0.001
41–52	1791 (15.0%)	1721 (13.6%)	2023 (17.7%)	1966 (18.8%)	
>52	2444 (20.5%)	2866 (22.6%)	2877 (25.1%)	2545 (24.3%)	
Employment type					
Permanent	5348 (44.9%)	6689 (52.7%)	5798 (50.6%)	5808 (55.5%)	<0.001
Temporary/daily	2043 (17.2%)	2127 (16.8%)	1748 (15.3%)	1197 (11.4%)	
Self-employed	4003 (33.6%)	3462 (27.3%)	3652 (31.9%)	3278 (31.3%)	
Others	510 (4.3%)	417 (3.3%)	254 (2.2%)	191 (1.8%)	
Shift work					
No	11,071 (93.0%)	11,741 (92.5%)	10,665 (93.1%)	9725 (92.8%)	0.232
Yes	833 (7.0%)	954 (7.5%)	787 (6.9%)	749 (7.2%)	
Job stress					
Low	4105 (34.5%)	3204 (25.2%)	2466 (21.5%)	1682 (16.1%)	<0.001
Middle	4897 (41.1%)	6142 (48.4%)	5417 (47.3%)	4994 (47.7%)	
High	2902 (24.4%)	3349 (26.4%)	3569 (31.2%)	3798 (36.3%)	
Job satisfaction					
Low	2143 (18.0%)	1938 (15.3%)	2099 (18.3%)	1809 (17.3%)	<0.001
High	9761 (82.0%)	10,757 (84.7%)	9353 (81.7%)	8665 (82.7%)	
Facing angry customers					
Rarely	10,452 (87.8%)	10,682 (84.1%)	9424 (82.3%)	8732 (83.4%)	<0.001
Sometimes	1092 (9.2%)	1540 (12.1%)	1578 (13.8%)	1370 (13.1%)	
Always	360 (3.0%)	473 (3.7%)	450 (3.9%)	372 (3.6%)	

^a^ ANOVA or Chi-squared test.

**Table 3 ijerph-19-10051-t003:** Association of automation anxiety and insomnia by logistic regression models. Bold indicates statistically significant values. [OR: odds ratio; CI: confidence interval].

	**Model A**			**Model B**		
	**OR**	**95% CI**	***p* Value**	**OR**	**95% CI**	***p* Value**
Automation anxiety (categorical)						
Q2 (Lower middle)	**1.19**	**1.08–1.31**	**0.004**	1.24	0.96–1.60	0.102
Q3 (Higher middle)	**1.27**	**1.15–1.41**	**<0.001**	**1.39**	**1.07–1.79**	**0.013**
Q4 (Highest)	**1.40**	**1.27–1.55**	**<0.001**	**1.98**	**1.55–2.53**	**<0.001**
Interaction terms						
Q2 × middle-aged				1.06	0.78–1.42	0.724
Q3 × middle-aged				1.02	0.75–1.38	0.884
Q4 × middle-aged				**0.74**	**0.55–0.98**	**0.037**
Q2 × old				0.88	0.65–1.18	0.386
Q3 × old				0.81	0.60–1.09	0.169
Q4 × old				**0.59**	**0.44–0.79**	**0.004**
	**Model C**			**Model D**		
	**OR**	**95% CI**	***p* value**	**OR**	**95% CI**	***p* Value**
Automation anxiety (continuous)	**1.03**	**1.02–1.04**	**<0.001**	**1.07**	**1.05–1.10**	**<0.001**
Interaction terms						
Automation anxiety × middle-aged				**0.96**	**0.94–0.99**	**0.006**
Automation anxiety × old				**0.95**	**0.92–0.97**	**0.001**

Model A: adjusted for age groups, gender, education, monthly income, occupations, working hours, employment type, shift work, job stress, job satisfaction, and facing angry customers (fully-adjusted model). Model B: Model A + interaction terms. Model C: fully adjusted model with automation anxiety as a continuous variable. Model D: Model C + interaction terms.

**Table 4 ijerph-19-10051-t004:** Stratified analyses by each age subgroup in a fully-adjusted model. Bold indicates statistically significant values. [OR: odds ratio; CI: confidence interval].

	Young (≤35 Years)			Middle-Aged (36–55 Years)			Old (>55 Years)		
	OR	95% CI	*p* Value	OR	95% CI	*p* Value	OR	95% CI	*p* Value
Model E *									
Automation anxiety (categorical)									
Q2 (Lower middle)	1.29	0.99–1.68	0.060	**1.29**	**1.11–1.50**	**0.001**	1.10	0.95–1.27	0.187
Q3 (Higher middle)	**1.42**	**1.09–1.85**	**0.011**	**1.39**	**1.19–1.62**	**<0.001**	**1.19**	**1.02–1.38**	**0.025**
Q4 (Highest)	**1.96**	**1.52–2.53**	**<0.001**	**1.40**	**1.20–1.64**	**<0.001**	**1.29**	**1.10–1.51**	**0.002**
Model F *									
Automation anxiety (continuous)	**1.07**	**1.05–1.10**	**<0.001**	**1.03**	**1.02–1.04**	**<0.001**	**1.02**	**1.01–1.04**	**0.002**

* Models adjusted for gender, education, monthly income, occupations, working hours, employment type, shift work, job stress, job satisfaction, and facing angry customers.

## Data Availability

The raw KWCS data is openly published online (https://www.kosha.or.kr/eoshri/index.do, accessed on 1 July 2022).

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
