# Peer review of "Association between Workers’ Anxiety over Technological Automation and Sleep Disturbance: Results from a Nationally Representative Survey"

_ijerph, 2022, doi:10.3390/ijerph191610051_

Round 1

Reviewer 1 Report

Dear author,

congratulations on your work which correlates automation anxiety and sleep disturbances. Your sample is impressive and I found the paper well written. 

I only suggest some minor improvements.

Introduction. 

Line 38 and following: you should be more precise about the health outcomes. You mention also physical health, so you have to define how automation can be harmful. Generally automation should reduce work-related diseases and not increase them, so you should clarify this point in the Introduction.

Line 51 and following: please provide if available any reference for the statement about young workers. Usually young workers accept better technology, you should argument better this concept.

In Introduction you discuss about the fact that automation will lead to a decrease in the number of workers employed, in the industry and in the third sector. Do you have any real life data about that?

Discussion

lines 220-229: as in introduction, what is the correlation between poor physical health and automation? please explain

Also I believe that the practical implication fo your study should be improved (automation is a ongoing and unstoppable process, so what can be the possible solutions to preserve workers'mental health?)

Author Response

Authors thank to the editor and reviewers for their valuable feedback on our submission to IJERPH and giving us the opportunity to submit the revised manuscript “Association between workers’ anxiety over technological automation and sleep disturbance: Results from a nationally representative survey”. Your insightful comments were very helpful in reshaping the manuscript. Please see our point-by-point response to each concern. (Please see the attachment)

Reviewer 2 Report

The article concerns an important issue which is the influence of working conditions on the human body. In this case, the impact of automation on sleep disturbances, anxiety and employee fears.

My insights to be developed:

Abstract

The Authors should expand on the last sentence. What policies are designed to alleviate employees' excessive anxiety.

1. Introduction

- lines 41-42 - Anxiety over automation is related to decreased job security, with is known to induce adverse mental health outcomes of workers - this sentence requires clarification. How is automation supposed to reduce work safety? Does it reduce work safety? What are the risks of implementing automation?

- lines 47 - what do the Authors understand by the concept of automation risk? - it would be worth explaining, what the Authors mean.

- lines 51-53 - the sentence should apply to the entire group of working people. Not only for young people, because other employees also play an important role in the implementation of the production process. Regarding employees, it would be worth explaining the age range of young employees, older employees. (This information appears later in the paper – lines 112-113. Is this age range also adopted for this part of the study?)

- in the introduction, it would be worth mentioning the psychosocial factors that only appear in the research results (lines 148-150).

The introduction does not show the positive sides of automation. I am also missing a reference to what can be done to prevent the negative health effects of introduced innovations in enterprises. In the last sentence, the authors refer to the Fourth Industrial Revolution (Industry 4.0) - it would be worth developing this topic in terms of occupational safety.

2. Materials and Methods

- lines 78-79 - The Authors excluded 3968 participants with missing data. The Authors should be explain what specifically contributed to the exclusion of these participants from the study.

- what is the relationship between the questions asked in the survey and automation and its impact on the employee? It would be worth clarifying that the questions concerned employees, e.g. production workers.

1) question 1 - do you experience stress in your work? (very general question)

2) question 2 - are you satisfied with your work?

3) question 3 - does your job involve with angry clients?

The section on the statistical tests used requires a justification for their selection. Authors should explain what the OR is, and how it is interpreted.

- in the chapter 2.4 - should also include information on gender (which proportion of the respondents were men and which women) - this information appears in Table 1.

3. Results

-  in the text, it would be worth analyzing the numerical data presented in the Tables

-  the lines 201-202 are empty

4. Discussion

- lines 220-222 - which research results are the authors writing about?

5. The conclusion needs to be expanded

I hope that my insights will be useful for the Authors of the study.

Author Response

Authors thank to the editor and reviewers for their valuable feedback on our submission to IJERPH and giving us the opportunity to submit the revised manuscript “Association between workers’ anxiety over technological automation and sleep disturbance: Results from a nationally representative survey”. Your insightful comments were very helpful in reshaping the manuscript. Please see our point-by-point response to each concern. (Please see the attachment )

Round 2

Reviewer 2 Report

I thank the Authors for their extensive explanations. The introduced changes made the text easier to read. They also allow for a better understanding of it. However, I encourage the Authors to check the study in terms of editing.